# Generalized Čebyšev and Grüss Type Results in Weighted Lebesgue Spaces

**Saad Ihsan Butt [1], Josip Pečarić [2] and Sanja Tipurić-Spužević [3],***

[1] Department of Mathematics, COMSATS University Islamabad, Lahore Campus, Punjab 54000, Pakistan; saadihsanbutt@cuilahore.edu.pk or saadihsanbutt@gmail.com
[2] Croatian Academy of Sciences and Arts, 1000 Zagreb, Croatia; pecaric@element.hr
[3] Faculty of Chemistry and Technology, University of Split, Rudera Boškovića 35, 21000 Split, Croatia
* Correspondence: stspuzevic@ktf-split.hr

**Abstract:** The classical Grüss and related inequalities have spurred a range of improvements, refinements, generalizations, and extensions. In the present article, we provide generalizations of Sokolov's inequality in weighted Lebesgue $L_\omega(\Omega, \mathcal{A}, \mu)$ spaces by employing the weighted Sonin's identity and Čebyšev functional. As a result, we provide a generalized Grüss inequality in which the bounding constants are improved with bounding functions in $L_\omega^p(\Omega, \mathcal{A}, \mu)$ spaces. As an application, we provide several new bounds for Jensen–Grüss type differences.

**Keywords:** Sonin's identity; Korkine's identity; Čebyšev functional; Grüss inequality; Jensen–Grüss inequality

**MSC:** 26D15; 26D20; 26D99

## 1. Introduction

Integral inequalities have been widely implemented in different fields of sciences. They are quite beneficial for developing and progressing the theory of functional analysis, differential equations, and applied numerical analysis by estimating sharp quadrature bounds. In the last two decades, various types of integral inequalities have been utilized in approximation theory and numerical analysis, enabling us to obtain better estimates by reducing the approximation of error involved. Integral inequalities provide explicit bounds pertaining to unknown functions. Integral inequalities serve as a necessary tool in the study of various classes of differential and integral equations (see [1–5] and references therein).

Mathematicians have placed effort in the development of inequality theory to find and explore a large variety of results that are fruitful and notable for applications. Now, inequalities have evolved to attain magnificent theoretical and applied usage in the fields of science and engineering. In inequality theory, the Čebyšev inequality [6] (p. 197) or [7] (p. 240) is renowned for synchronous functions that produce limit values and helps to generate a variety of new inequalities. There exists a huge sequence of complements to the Čebyšev inequality which provides estimates for Čebyšev quotients and differences in the form of Grüss and Grüss-type inequalities [6] (p. 43). In diverse field of research, these inequalities have an immense number of variants with several applications in statistical problems, probability, fractional calculus, and numerical quadrature formulas (see [8–12]).

For two Lebesgue integrable functions h, g : $[u, v] \to \mathbb{R}$, the Čebyšev functional is given by

$$\mathfrak{C}(h, g) = \frac{1}{u-v} \int_u^v h(\zeta)g(\zeta)d\zeta - \frac{1}{(u-v)^2}\left(\int_u^v h(\zeta)d\zeta\right)\left(\int_u^v g(\zeta)d\zeta\right)$$

In 1882, Čebyšev proved in [13] that

$$|\mathfrak{C}(h, \mathfrak{g})| \leq \frac{1}{12}(u - v)^2 \|h'\|_\infty \|\mathfrak{g}'\|_\infty$$

where $h, \mathfrak{g} : [u, v] \to \mathbb{R}$ are such that $h', \mathfrak{g}'$ are continuous on $[u, v]$ and $\|h'\|_\infty = \sup\limits_{t \in [u,v]} |h'(t)|$.

In their 1934 remarkable paper [14], Grüss proved that

$$|\mathfrak{C}(h, \mathfrak{g})| \leq \frac{1}{4}(M - m)(\Gamma - \gamma),$$

provided that there exist the real numbers $m, M, \gamma, \Gamma$ such that

$$m \leq h(\zeta) \leq M, \quad \gamma \leq \mathfrak{g}(\zeta) \leq \Gamma$$

for a.e. $\zeta \in [u, v]$. The constant $1/4$ is the best possible.

In 1963, Sokolov in [15] proved that

$$|\mathfrak{C}(h, \mathfrak{g})| \leq \frac{M - m}{2(u - v)} \int_u^v \left| \left( \mathfrak{g}(\zeta) - \frac{1}{(u - v)} \int_u^v \mathfrak{g}(t)dt \right) \right| d\zeta$$

provided that there exist the real numbers $m, M$, such that

$$m \leq h(\zeta) \leq M$$

for a.e. $\zeta \in [u, v]$. The constant $1/2$ is the best possible.

These are the most demanding inequalities in computational and applied mathematics due to their effective and immense applications in perturbed quadrature rules [16,17] and approximation of integral transforms [18,19].

Throughout the paper, we use $1 \leq p, q \leq \infty$ as conjugate exponents that are $\frac{1}{q} + \frac{1}{p} = 1$. Consider the space of $p$-power integrable functions $L_{[u,v]}^p$ with norm

$$\|h\|_p = \left( \int_u^v |h(\zeta)|^p d\zeta \right)^{\frac{1}{p}}$$

and the space $L_{[u,v]}^\infty$ with the norm

$$\|h\|_\infty = ess \sup\limits_{\zeta \in [u,v]} |h(\zeta)|.$$

In [20], M. Niezgoda recently provided an extended Grüss theorem for certain classes of bounding functions instead of bounding constants as follows:

**Theorem 1.** *Consider the functions* $h, \mathfrak{g}, \eta, \kappa, \gamma, \vartheta \in L_{[u,v]}^2$ *such that*

*(i)*   $\eta + \kappa$ *and* $\gamma + \vartheta$ *are constant functions.*
*(ii)*  $\eta(\zeta) \leq h(\zeta) \leq \kappa(\zeta)$ *and* $\gamma(\zeta) \leq \mathfrak{g}(\zeta) \leq \vartheta(\zeta)$ *for all* $\zeta \in [u, v]$ *or more generally*

$$\int_u^v (\kappa(\zeta) - h(\zeta))(h(\zeta) - \eta(\zeta))d\zeta \geq 0 \quad and \quad \int_u^v (\vartheta(\zeta) - \mathfrak{g}(\zeta))(\mathfrak{g}(\zeta) - \gamma(\zeta))d\zeta \geq 0.$$

*Then, we have the inequality*

$$|\mathfrak{C}(h, \mathfrak{g})| \leq \frac{1}{4(u - v)} \|\kappa - \eta\|_2 \|\vartheta - \gamma\|_2.$$

In [9], M. Niezgoda also obtained generalization of the above result for $L^p$-spaces.

**Theorem 2.** *Consider the functions* $h, \eta, \kappa \in L^p_{[u,v]}$, *and* $\mathfrak{g} \in L^q_{[u,v]}$ *such that*

*(i)*   $\eta + \kappa$ *is a constant function.*
*(ii)*  $\eta(\zeta) \leq h(\zeta) \leq \kappa(\zeta)$ *for all* $\zeta \in [u,v]$.

*Then, we have the inequality*

$$|\mathfrak{C}(h,\mathfrak{g})| \leq \frac{1}{2(u-v)} \|\kappa - \eta\|_p \left\| \mathfrak{g} - \frac{1}{u-v} \int_u^v \mathfrak{g}(s)ds \right\|_q. \tag{1}$$

It is interesting to note that as a special case of $p = q = 2$, Theorem 1 is the direct consequence of Theorem 2.

Recently, M. Niezgoda [21] investigated pre-Grüss-type inequalities pertaining to continuous functions possessing only one point of non-differentiability. For $x_0 \in [u,v]$, let $\mathcal{D}(\zeta_0)$ be the class of all continuous functions $h : [u,v] \to \mathbb{R}$ differentiable on the set $\langle u, \zeta_0 \rangle \cup \langle \zeta_0, v \rangle$ and such that

$$M_l = \sup_{\zeta \in \langle u, \zeta_0 \rangle} |h'(\zeta)| < \infty \text{ and } M_r = \sup_{\zeta \in \langle \zeta_0, v \rangle} |h'(\zeta)| < \infty. \tag{2}$$

In case $\zeta_0 = u$ (resp. $\zeta_0 = v$) we set $M_l = 0$ (resp $M_r = 0$).

For given function $h \in \mathcal{D}(\zeta_0)$ we define

$$\eta_{h,\zeta_0}(\zeta) = \begin{cases} M_l(\zeta - \zeta_0) + h(\zeta_0) \text{ for } \zeta \in [u, \zeta_0], \\ -M_r(\zeta - \zeta_0) + h(\zeta_0) \text{ for } \zeta \in [\zeta_0, v] \end{cases} \tag{3}$$

and

$$\kappa_{h,\zeta_0}(\zeta) = \begin{cases} -M_l(\zeta - \zeta_0) + h(\zeta_0) \text{ for } \zeta \in [u, \zeta_0], \\ M_r(\zeta - \zeta_0) + h(\zeta_0) \text{ for } \zeta \in [\zeta_0, v] \end{cases} \tag{4}$$

where $M_l$ and $M_r$ are defined by (2).

In [21], M. Niezgoda established the following notable result:

**Lemma 1.** *Let* $h \in \mathcal{D}(\zeta_0)$ *for some* $\zeta_0 \in [u,v]$. *Denote* $\eta = \eta_{h,\zeta_0}(\zeta)$ *and* $\kappa = \kappa_{h,\zeta_0}(\zeta)$, *where* $M_l$ *and* $M_r$ *are defined by* (2). *Then, we have that*

*(i)*   $\eta + \kappa$ *is a constant function.*
*(ii)*  $\eta(\zeta) \leq h(\zeta) \leq \kappa(\zeta)$ *for all* $\zeta \in [u,v]$.
*(iii)* *The* $L_p$-*norm of* $\kappa - \eta$ *is*

$$\|\kappa - \eta\|_p = \begin{cases} \frac{2}{(p+1)^{1/p}} \left[ M_l^p(\zeta_0 - u)^{p+1} + M_r^p(v - \zeta_0)^{p+1} \right]^{1/p} \text{ for } 1 \leq p < \infty, \\ 2 \max\{ M_l(\zeta_0 - u), M_r(v - \zeta_0) \} \text{ for } p = \infty. \end{cases}$$

Later, in [22], Aljinović et al. provided new weighted estimates of the Grüss inequality with the bounding functions in weighted $L^p_{\omega,[u,vs.]}$ spaces by considering uniform weight functions. As a result, a new generalized variant of the Ostrwoski inequality and some applications to weighted quadrature formulae were provided as well.

Motivated by the above literature review, the aim of this paper is to present generalizations of Grüss- and Sokolov-type inequalities in weighted Lebesgue $L_w(\Omega, A, \mu)$ spaces by using the weighted Sonin's identity. To begin, we need to introduce the basic notions. Let $(\Omega, \mathcal{A}, \mu)$ be a measurable space, and for the $\mu$-measurable function $\omega : \Omega \to \mathbb{R}$, with $\omega(\zeta) > 0$ for $\mu$-a.e. $\zeta \in \Omega$, consider the Lebesgue space

$$L_\omega(\Omega, \mathcal{A}, \mu) = \left\{ h : \Omega \to \mathbb{R}, h \text{ is } \mu - \text{measurable and } \int_\Omega \omega(\zeta)|h(\zeta)|d\mu(\zeta) < \infty \right\}.$$

For $h, \mathfrak{g} : \Omega \to \mathbb{R}$ are $\mu$-measurable functions and $h, \mathfrak{g}, h\mathfrak{g} \in L_\omega(\Omega, \mathcal{A}, \mu)$; then, the weighted Čebyšev functional is

$$\mathfrak{C}_\omega(h, \mathfrak{g}) = \frac{1}{W} \int_\Omega \omega(\zeta) h(\zeta) \mathfrak{g}(\zeta) d\mu(\zeta) - \left( \frac{1}{W} \int_\Omega \omega(\zeta) h(\zeta) d\mu(\zeta) \right) \left( \frac{1}{W} \int_\Omega \omega(\zeta) \mathfrak{g}(\zeta) d\mu(\zeta) \right) \tag{5}$$

where $W = \int_\Omega \omega(\zeta) d\mu(\zeta) > 0$.

We will apply Sonin's identity (see [23])

$$\mathfrak{C}_\omega(h, \mathfrak{g}) = \int_u^v \omega(\zeta) (h(\zeta) - \chi) \left( \mathfrak{g}(\zeta) - \int_u^v \omega(\zeta) \mathfrak{g}(\zeta) d\zeta \right) d\mu(\zeta)$$

where $\chi$ is an arbitrary real number and $\omega$ is the normalized weight function, that is, $W = \int_u^v w d\zeta = 1$.

Let us note that the following identity is a generalization of *Sonin's identity*

$$\mathfrak{C}_\omega(h, \mathfrak{g}) = \frac{1}{W} \int_\Omega \omega(\zeta) (h(\zeta) - \chi)(\mathfrak{g}(\zeta) - \bar{\mathfrak{g}}) d\mu(\zeta) \tag{6}$$

where

$$\bar{\mathfrak{g}} = \frac{1}{W} \int_\Omega \omega(\zeta) \mathfrak{g}(\zeta) d\mu(\zeta). \tag{7}$$

## 2. Grüss- and Sokolov-Type Inequalities for $L_\omega(\Omega, \mathcal{A}, \mu)$ Spaces

Here and hereafter, $L_\omega^p(\Omega, \mathcal{A}, \mu)$ denotes the function space $L_\omega(\Omega, \mathcal{A}, \mu)$ equipped with the norm

$$\|h\|_{\omega, p} = \left( \int_\Omega \omega(\zeta)(h(\zeta))^p d\mu(\zeta) \right)^{1/p} = \|h\|_{\omega, \Omega, p}, \ \|h\|_{\omega, \infty} = ess \sup_{z \in \Omega} |h(\zeta)| = \|h\|_{\omega, \Omega, \infty}$$

and $\mathfrak{g} \in (\Omega(\mathbb{R}), \mu)$ indicates that $\mathfrak{g} : \Omega \to \mathbb{R}$ is $\mu$-measurable function.

The following result is a simple consequence of Sonin's identity (6).

**Theorem 3.** *Let* $h, \mathfrak{g} \in (\Omega(\mathbb{R}), \mu)$ *and* $h, \mathfrak{g}, h\mathfrak{g} \in L_\omega(\Omega, A, \mu)$. *Then, we have*

$$|\mathfrak{C}_\omega(h, \mathfrak{g})| \leq \frac{1}{W} \int_\Omega \omega(\zeta) |(h(\zeta) - \chi)| |(\mathfrak{g}(\zeta) - \bar{\mathfrak{g}})| d\mu(\zeta) \tag{8}$$

*where* $\bar{\mathfrak{g}}$ *is weighted average given in* (7).

The following result is a direct consequence of Theorem 3.

**Theorem 4.** *Let* $h, \mathfrak{g}, c \in (\Omega(\mathbb{R}), \mu)$ *such that* $h, \mathfrak{g}, h\mathfrak{g} \in L_\omega(\Omega, A, \mu)$. *If*

$$|h(\zeta) - \chi| \leq c(\zeta) \text{ for } \mu - a.e. \ \zeta \in \Omega. \tag{9}$$

*Then, we have the inequality*

$$|\mathfrak{C}_\omega(h, \mathfrak{g})| \leq \frac{1}{W} \int_\Omega \omega(\zeta) c(\zeta) |(\mathfrak{g}(\zeta) - \bar{\mathfrak{g}})| d\mu(\zeta). \tag{10}$$

**Proof.** We obtain (10) from (8) and (9). □

**Theorem 5.** *Let* $h, g \in (\Omega(\mathbb{R}), \mu)$ *such that* $h, g, hg \in L_\omega(\Omega, A, \mu)$ *and* $-\infty < m \leq M < +\infty$ *such that*

$$m \leq h(\zeta) \leq M \text{ for } \mu - a.e. \, \zeta \in \Omega. \tag{11}$$

*Then, we have the inequality*

$$|\mathfrak{C}_\omega(h, g)| \leq \frac{M - m}{2W} \int_\Omega \omega(\zeta) |(g(\zeta) - \bar{g})| d\mu(\zeta). \tag{12}$$

**Proof.** If (11) is valid, then we have

$$\left| h(\zeta) - \frac{m + M}{2} \right| \leq \frac{M - m}{2}$$

for $\mu$-a.e. $\zeta \in \Omega$. Thus, for $\chi = \frac{m+M}{2}$ and $c(\zeta) = \frac{M-m}{2}$, we obtain (12) from (10). □

**Remark 1.** *The above result was first proven by I. G. Sokolov [15] for the case of* $\Omega = [u, v], \omega(\zeta) = 1, \mu(\zeta) = \zeta$. *The same result was rediscovered by X. L. Cheng and J. Sun [17] without weights. Additionally, the above generalizations were proven by P. Cerone and S. S. Dragomir [16]; here, however, we employ Sonin's identity to obtain our results.*

**Remark 2.** *Let us note that Theorem 5 is an improvement of the well-known Grüss inequality*

$$|\mathfrak{C}_\omega(h, g)| \leq \frac{1}{4}(M - m)(\Gamma - \gamma)$$

*where* (11) *and*

$$-\infty < \gamma \leq g(\zeta) \leq \Gamma < +\infty \text{ for } \mu - a.e. \zeta \in \Omega \tag{13}$$

*is valid.*

In fact, the following result from [24] is valid

$$|\mathfrak{C}_\omega(h, g)| \leq \frac{1}{2}(M - m) \frac{1}{W} \int_\Omega \omega(\zeta) |(g(\zeta) - \bar{g})| d\mu(\zeta)$$

$$\leq \frac{1}{2}(M - m) \frac{1}{W^2} \int_\Omega \int_\Omega \omega(\zeta) \omega(t) |(g(\zeta) - g(t))| d\mu(\zeta) d\mu(t)$$

$$\leq \frac{1}{2}(M - m)(\mathfrak{C}_\omega(g, g))^{\frac{1}{2}}$$

$$\leq \frac{1}{4}(M - m)(\Gamma - \gamma).$$

If (9) is valid, then for $p \geq 1, \frac{1}{p} + \frac{1}{q} = 1, c \in L^p_\omega(\Omega, \mathcal{A}, \mu)$ and $g \in L^q_\omega(\Omega, \mathcal{A}, \mu)$

$$|\mathfrak{C}_\omega(h, g)| = \frac{1}{W} \int_\Omega \omega(\zeta) |c(\zeta)| |(g(\zeta) - \bar{g})| d\mu(\zeta)$$

$$\leq \left( \frac{1}{W} \int_\Omega \omega(\zeta)(c(\zeta))^p d\mu(\zeta) \right)^{\frac{1}{p}} \left( \frac{1}{W} \int_\Omega \omega(\zeta) |(g(\zeta) - \bar{g})|^q d\mu(\zeta) \right)^{\frac{1}{q}} \tag{14}$$

$$= \frac{1}{W} \|c\|_{\omega, \Omega, p} \|g - \bar{g}\|_{\omega, \Omega, q}.$$

If (11) is valid for $\chi = \frac{m+M}{2}$, we have the following result from [25]

$$|\mathfrak{C}_\omega(h, \mathfrak{g})| \le \frac{M - m}{2} \frac{1}{W} \int_\Omega \omega(\zeta) |(\mathfrak{g}(\zeta) - \bar{\mathfrak{g}})| d\mu(\zeta)$$

$$\le \frac{M - m}{2} \left( \frac{1}{W} \int_\Omega \omega(\zeta) |(\mathfrak{g}(\zeta) - \bar{\mathfrak{g}})|^q d\mu(\zeta) \right)^{\frac{1}{q}}$$

$$\le \frac{M - m}{2W} \|\mathfrak{g} - \bar{\mathfrak{g}}\|_{\omega, \Omega, q}.$$

**Corollary 1.** *Let* h, $\mathfrak{g}$ *be as defined in Theorem 5. In addition, let* $\Omega = \bigcup_{i=1}^{n} \Omega_i$ *and* $\bigcap_{i=1}^{n} \Omega_i = \varnothing$ *be such that* $c_i \in (\Omega_i(\mathbb{R}), \mu),$ h. *If*

$$|h(\zeta) - \chi_i| \le c_i(\zeta) \text{ for } \mu - a.e. \ \zeta \in \Omega_i,$$

*then we have the inequality*

$$|\mathfrak{C}_\omega(h, \mathfrak{g})| \le \frac{1}{W} \sum_{i=1}^{n} \int_{\Omega_i} \omega(\zeta) c_i(\zeta) |(\mathfrak{g}(\zeta) - \bar{\mathfrak{g}})| d\mu(\zeta). \tag{15}$$

**Theorem 6.** *Let the functions* h, $\eta, \kappa \in L_\omega^p(\Omega, \mathcal{A}, \mu)$ *and* $\mathfrak{g} \in L_\omega^q(\Omega, \mathcal{A}, \mu)$ *with* $1 \le p, q \le \infty$ *be conjugate exponents such that*

*(i)* $\eta + \kappa$ *is a constant function.*
*(ii)* $\eta(\zeta) \le h(\zeta) \le \kappa(\zeta)$ *for all* $\mu - a.e. \zeta \in \Omega$. *Then, we have the inequality*

$$|\mathfrak{C}_\omega(h, \mathfrak{g})| \le \frac{1}{2W} \|\kappa - \eta\|_{\omega, \Omega, p} \|\mathfrak{g} - \bar{\mathfrak{g}}\|_{\omega, \Omega, q}. \tag{16}$$

**Proof.** If *(ii)* is valid, we have

$$\left| h(\zeta) - \frac{\eta(\zeta) + \kappa(\zeta)}{2} \right| \le \frac{\kappa(\zeta) - \eta(\zeta)}{2}$$

for $\mu$-a.e.$\zeta \in \Omega$. Thus, for $\chi = \frac{\eta(\zeta) + \kappa(\zeta)}{2}$ and $c(\zeta) = \frac{\kappa(\zeta) - \eta(\zeta)}{2}$, we obtain (16) from (9) and (14). $\square$

**Lemma 2.** *Let* $\Omega = [u, v]$ *and* h $\in \mathcal{D}(\zeta_0)$ *for some* $\zeta_0 \in [u, v]$. *Denote* $\eta = \eta_{h, x_0}(\zeta)$ *and* $\kappa = \kappa_{h, \zeta_0}(\zeta)$, *where* $M_l$ *and* $M_r$ *are defined by* (2). *Then, we have that*

*(i)* $\eta + \kappa$ *is a constant function.*
*(ii)* $\eta(\zeta) \le h(\zeta) \le \kappa(\zeta)$ *for all* $\zeta \in [u, v]$.
*(iii)* *The* $L_w^p$-*norm of* $\kappa - \eta$ *is given by*

$$\|\kappa - \eta\|_{\omega, p} = \begin{cases} 2 \left[ M_l^p \int_u^{\zeta_0} \omega(\zeta)(\zeta_0 - \zeta)^p d\mu(\zeta) + M_r^p \int_{\zeta_0}^{v} \omega(\zeta)(\zeta - \zeta_0)^p d\mu(\zeta) \right]^{1/p} & \text{for } 1 \le p < \infty, \\ 2 \max\{M_l(\zeta_0 - u), M_r(v - \zeta_0)\} & \text{for } p = \infty. \end{cases} \tag{17}$$

**Proof.** First, we take $\zeta_0 \in \langle u, v \rangle$. From (3) and (4), we find that $\eta(\zeta) + \kappa(\zeta) = 2h(\zeta_0)$ which proves *(i)*.

To prove *(ii)*, we will assume first that $x \in [u, \zeta_0]$. By Lagrange's mean value theorem, for all $x \in [u, \zeta_0)$, exists $\xi_1 \in \langle \zeta, \zeta_0 \rangle$ such that

$$h(\zeta) - h(\zeta_0) = h'(\xi_1)(\zeta - \zeta_0).$$

It is clear that

$$-M_l \leq -\left|h'(\xi_1)\right| \leq h'(\xi_1) \leq \left|h'(\xi_1)\right| \leq M_l$$

and $\zeta - \zeta_0 < 0$ which gives

$$-M_l(\zeta - \zeta_0) + h(\zeta_0) \geq h'(\xi_1)(\zeta - \zeta_0) + h(\zeta_0) \geq M_l(\zeta - \zeta_0) + h(\zeta_0). \tag{18}$$

Equation (18) is also valid for $\zeta = \zeta_0$. Then, from (3)–(4) we have

$$\eta(\zeta) \leq h(\zeta) \leq \kappa(\zeta) \text{ for all } \zeta \in [u, \zeta_0]. \tag{19}$$

Similarly, for $\zeta \in \langle \zeta_0, v]$ we have

$$h(\zeta) - h(\zeta_0) = h'(\xi_2)(\zeta - \zeta_0)$$

for some $\xi_2 \in \langle \zeta_0, \zeta \rangle$. Then, for $\zeta \in \langle \zeta_0, v]$ we have

$$-M_r(\zeta - \zeta_0) + h(\zeta_0) \leq h'(\xi_2)(\zeta - \zeta_0) + h(\zeta_0) \leq M_r(\zeta - \zeta_0) + h(\zeta_0). \tag{20}$$

Equation (20) is also valid for $\zeta = \zeta_0$. From that, we have

$$\eta(\zeta) \leq h(\zeta) \leq \kappa(\zeta) \text{ for all } \zeta \in [\zeta_0, v]. \tag{21}$$

From (19) and (21), we obtain (*ii*).

(*iii*) Simple calculation for $1 \leq p < \infty$ gives

$$\|\kappa - \eta\|_{\omega, \Omega, p} = \left( \int_u^v \omega(\zeta) \left( \kappa_{h, \zeta_0}(\zeta) - \eta_{h, \zeta_0}(\zeta) \right)^p d\mu(\zeta) \right)^{1/p}$$

$$= 2 \left[ M_l^p \int_u^{\zeta_0} \omega(\zeta)(\zeta_0 - \zeta)^p d\mu(\zeta) + M_r^p \int_{\zeta_0}^v \omega(\zeta)(\zeta - \zeta_0)^p d\mu(\zeta) \right]^{1/p}$$

and for $p = \infty$

$$\|\kappa - \eta\|_{\omega, \Omega, \infty} = ess \sup_{\zeta \in [u,v]} \left| \kappa_{h, \zeta_0}(\zeta) - \eta_{h, \zeta_0}(\zeta) \right|$$

$$= 2 \max\{M_l(\zeta_0 - u), M_r(v - \zeta_0)\}.$$

Here (*i*), (*ii*), and (*iii*) can be determined with similar steps to the cases $\zeta_0 = u$ (when $M_l = 0$ and $M_r = \sup_{\zeta \in \langle u,v \rangle} |h'(\zeta)|$) and $\zeta_0 = v$ (when $M_r = 0$ and $M_l = \sup_{\zeta \in \langle u,v \rangle} |h'(\zeta)|$). □

**Remark 3.** *From Lemma 2 we obtain Lemma 1 for* $\omega(\zeta) = 1$ *and* $\mu(\zeta) = \zeta$.

**Corollary 2.** *With the same assumptions as in Lemma 2 we obtain*

$$|\mathfrak{C}_\omega(h, \mathfrak{g})| \leq \frac{1}{W} \left[ M_l^p \int_u^{\zeta_0} \omega(\zeta)(\zeta_0 - \zeta)^p d\mu(\zeta) + M_r^p \int_{\zeta_0}^v \omega(\zeta)(\zeta - \zeta_0)^p d\mu(\zeta) \right]^{1/p}$$

$$\times \|\mathfrak{g} - \bar{\mathfrak{g}}\|_{\omega, q}, \text{ for } 1 \leq p < \infty$$

*and*

$$|\mathfrak{C}_\omega(h, \mathfrak{g})| \leq \frac{1}{W} \max\{M_l(\zeta_0 - u), M_r(v - \zeta_0)\} \|\mathfrak{g} - \bar{\mathfrak{g}}\|_{\omega, 1}, \text{ for } q = 1.$$

**Proof.** Because $h \in \mathcal{D}(\zeta_0)$, $h$ is continuous on $[u, v]$, it is implied that $h \in L^p_{\omega,[u,v]}$. Now, we simply employ Lemma 2 to Theorem 6 . $\square$

**Theorem 7.** *Let* $h, \mathfrak{g} \in (\Omega(\mathbb{R}), \mu)$ *and* $\mathfrak{g} \in L^q_\omega(\Omega, \mathcal{A}, \mu)$. *If* $\Omega = [u, v]$ *and* $\chi = h(\zeta_0)$. *Then, we have the inequalities*

$$|\mathfrak{C}_\omega(h, \mathfrak{g})| \leq \frac{1}{W} \left[ M_l^p \int_u^{\zeta_0} \omega(\zeta)(\zeta_0 - \zeta)^p d\mu(\zeta) + M_r^p \int_{\zeta_0}^v \omega(\zeta)(\zeta - \zeta_0)^p d\mu(\zeta) \right]^{1/p}$$
$$\times \|\mathfrak{g} - \overline{\mathfrak{g}}\|_{\omega,q} \ for \ 1 \leq p < \infty$$

*and*

$$|\mathfrak{C}_\omega(h, \mathfrak{g})| \leq \frac{1}{W} \max\{M_l(\zeta_0 - u), M_r(v - \zeta_0)\} \|\mathfrak{g} - \overline{\mathfrak{g}}\|_{\omega,1}, \ for \ p = \infty.$$

**Proof.** Inserting $\Omega = [u, v]$ and $\chi = h(\zeta_0)$ in (6) for $1 \leq p < \infty$, we obtain

$$\mathfrak{C}_\omega(h, \mathfrak{g}) = \frac{1}{W} \int_u^v \omega(\zeta)(h(\zeta) - h(\zeta_0))(\mathfrak{g}(\zeta) - \overline{\mathfrak{g}}) d\mu(\zeta). \tag{22}$$

By Lagrange's mean value theorem we have

$$|(h(\zeta) - h(\zeta_0))| \leq \begin{cases} M_l(\zeta_0 - \zeta), \ u \leq \zeta \leq \zeta_0, \\ M_r(\zeta - \zeta_0), \ \zeta_0 \leq \zeta \leq v. \end{cases} \tag{23}$$

From (22), we can obtain

$$\mathfrak{C}_\omega(h, \mathfrak{g}) = \frac{1}{W} \int_u^{\zeta_0} \omega(\zeta)(h(\zeta) - h(\zeta_0))(\mathfrak{g}(\zeta) - \overline{\mathfrak{g}}) d\mu(\zeta)$$
$$+ \frac{1}{W} \int_{\zeta_0}^v \omega(\zeta)(h(\zeta) - h(\zeta_0))(\mathfrak{g}(\zeta) - \overline{\mathfrak{g}}) d\mu(\zeta) \tag{24}$$

Then, by taking absolute value with the triangular and Hölder's inequalities, we obtain

$$|\mathfrak{C}_\omega(h, \mathfrak{g})| \leq \left| \frac{1}{W} \int_u^{\zeta_0} \omega(\zeta)(h(\zeta) - h(\zeta_0))(\mathfrak{g}(\zeta) - \overline{\mathfrak{g}}) d\mu(\zeta) \right|$$
$$+ \left| \frac{1}{W} \int_{\zeta_0}^v \omega(\zeta)(h(\zeta) - h(\zeta_0))(\mathfrak{g}(\zeta) - \overline{\mathfrak{g}}) d\mu(\zeta) \right|$$
$$\leq \frac{1}{W} \|h(\zeta) - h(\zeta_0)\|_{\omega,p,[u,\zeta_0]} \|\mathfrak{g} - \overline{\mathfrak{g}}\|_{\omega,q,[u,\zeta_0]}$$
$$+ \frac{1}{W} \|h(\zeta) - h(\zeta_0)\|_{\omega,p,[\zeta_0,v]} \|\mathfrak{g} - \overline{\mathfrak{g}}\|_{\omega,q,[\zeta_0,v]}. \tag{25}$$

Now, employing the bounds of (23) in (25) and considering the discrete Hölder's inequality leads us to:

$$\frac{1}{W}\|\mathrm{h}(\zeta) - \mathrm{h}(\zeta_0)\|_{\omega,p,[u,\zeta_0]}\|\mathfrak{g} - \overline{\mathfrak{g}}\|_{\omega,q,[u,\zeta_0]} + \frac{1}{W}\|\mathrm{h}(\zeta) - \mathrm{h}(\zeta_0)\|_{\omega,p,[\zeta_0,v]}\|\mathfrak{g} - \overline{\mathfrak{g}}\|_{\omega,q,[\zeta_0,v]}$$

$$\leq \frac{M_l}{W}\left(\int_u^{\zeta_0}\omega(\zeta)(\zeta_0 - \zeta)^p d\mu(\zeta)\right)^{\frac{1}{p}}\|\mathfrak{g} - \overline{\mathfrak{g}}\|_{\omega,q,[u,\zeta_0]}$$

$$+ \frac{M_r}{W}\left(\int_{\zeta_0}^{v}\omega(\zeta)(\zeta - \zeta_0)^p d\mu(\zeta)\right)^{\frac{1}{p}}\|\mathfrak{g} - \overline{\mathfrak{g}}\|_{\omega,q,[\zeta_0,v]}$$

$$= \frac{1}{W}\left(M_l^p\int_u^{\zeta_0}\omega(\zeta)(\zeta_0 - \zeta)^p d\mu(\zeta) + M_r^p\int_{\zeta_0}^{v}\omega(\zeta)(\zeta - \zeta_0)^p d\mu(\zeta)\right)^{\frac{1}{p}}$$

$$\cdot\left(\|\mathfrak{g} - \overline{\mathfrak{g}}\|_{\omega,q,[u,\zeta_0]} + \|\mathfrak{g} - \overline{\mathfrak{g}}\|_{\omega,q,[\zeta_0,v]}\right)$$

$$= \frac{1}{W}\left(M_l^p\int_u^{\zeta_0}\frac{1}{W}\omega(\zeta)(\zeta_0 - \zeta)^p d\mu(\zeta) + M_r^p\int_{\zeta_0}^{v}\frac{1}{W}\omega(\zeta)(\zeta - \zeta_0)^p d\mu(\zeta)\right)^{\frac{1}{p}}\|\mathfrak{g} - \overline{\mathfrak{g}}\|_{\omega,q}.$$

Similarly, for $p = \infty$ we obtain

$$|\mathfrak{C}_\omega(\mathrm{h}, \mathfrak{g})| \leq \frac{M_l}{W}\int_u^{\zeta_0}\omega(\zeta)(\zeta_0 - \zeta)|(\mathfrak{g}(\zeta) - \overline{\mathfrak{g}})|d\mu(\zeta)$$

$$+ \frac{M_r}{W}\int_{\zeta_0}^{v}\omega(\zeta)(\zeta - \zeta_0)|(\mathfrak{g}(\zeta) - \overline{\mathfrak{g}})|d\mu(\zeta)$$

$$\leq M_l(\zeta_0 - u)\frac{1}{W}\left(\int_u^{\zeta_0}\omega(\zeta)|(\mathfrak{g}(\zeta) - \overline{\mathfrak{g}})|d\mu(\zeta)\right)$$

$$+ M_r(v - \zeta_0)\frac{1}{W}\left(\int_{\zeta_0}^{v}\omega(\zeta)|(\mathfrak{g}(\zeta) - \overline{\mathfrak{g}})|d\mu(\zeta)\right)$$

$$\leq \frac{1}{W}\max\{M_l(\zeta_0 - u), M_r(v - \zeta_0)\}\left(\int_u^{v}\omega(\zeta)|(\mathfrak{g}(\zeta) - \overline{\mathfrak{g}})|d\mu(\zeta)\right).$$

$\square$

**Corollary 3.** *Let $\Omega = \bigcup\limits_{i=1}^{n}\Omega_i, \bigcap\limits_{i=1}^{n}\Omega_i = \varnothing$ and the functions $\eta_i, \kappa_i \in L_\omega^p(\Omega, \mathcal{A}, \mu), \mathrm{h}, \in L_\omega^p$ $(\Omega, \mathcal{A}, \mu)$ and $\mathfrak{g} \in L_\omega^q(\Omega, \mathcal{A}, \mu)$ with $1 \leq p, q \leq \infty$ be conjugate exponents such that*

*(i)    $\eta_i + \kappa_i$ are a constant functions, $i = 1, ..., n$*

*(ii)   $\eta_i(\zeta) \leq \mathrm{h}(\zeta) \leq \kappa_i(\zeta)$ for all $\mu - a.e.\zeta \in \Omega_i, \; i = 1, ..., n$. Then, we have the inequality*

$$2W|\mathfrak{C}_\omega(\mathrm{h}, \mathfrak{g})| \leq \left(\sum_{i=1}^{n}\|\eta_i - \kappa_i\|_{w,\Omega_i,p}^p\right)^{\frac{1}{p}}\|\mathfrak{g} - \overline{\mathfrak{g}}\|_{w,\Omega,q}. \tag{26}$$

**Proof.** This theorem is a simple consequence of Theorem 7. Specifically, from Theorem 7 and the application of the discrete Hölder's inequality, we obtain

$$
\begin{aligned}
2W|\mathfrak{C}_\omega(\mathrm{h},\mathfrak{g})| \;&\leq\; \int_{\Omega_1} \omega(\zeta)(\eta_1(\zeta)+\kappa_1(\zeta))|(\mathfrak{g}(\zeta)-\overline{\mathfrak{g}})|d\mu(\zeta) + \cdots \\
&+\; \int_{\Omega_n} \omega(\zeta)(\eta_n(\zeta)+\kappa_n(\zeta))|(\mathfrak{g}(\zeta)-\overline{\mathfrak{g}})|d\mu(\zeta) \\
&\leq\; \|\eta_1-\kappa_1\|_{w,\Omega_1,p}\|\mathfrak{g}-\overline{\mathfrak{g}}\|_{w,\Omega_1,q} + \cdots + \|\eta_n-\kappa_n\|_{w,\Omega_n,p}\|\mathfrak{g}-\overline{\mathfrak{g}}\|_{w,\Omega_n,q} \\
&\leq\; \left(\|\eta_1-\kappa_1\|_{w,\Omega_1,p}^p + \cdots + \|\eta_n-\kappa_n\|_{w,\Omega_n,p}^p\right)^{\frac{1}{p}}\left(\|\mathfrak{g}-\overline{\mathfrak{g}}\|_{w,\Omega_1,q}^q + \cdots + \|\mathfrak{g}-\overline{\mathfrak{g}}\|_{w,\Omega_n,q}^q\right)^{\frac{1}{q}} \\
&=\; \left(\sum_{i=1}^n \|\eta_i-\kappa_i\|_{w,\Omega_i,p}^p\right)^{\frac{1}{p}}\left(\sum_{i=1}^n \|\mathfrak{g}-\overline{\mathfrak{g}}\|_{w,\Omega_i,q}^q\right)^{\frac{1}{q}} \\
&=\; \left(\sum_{i=1}^n \|\eta_i-\kappa_i\|_{w,\Omega_i,p}^p\right)^{\frac{1}{p}}\|\mathfrak{g}-\overline{\mathfrak{g}}\|_{w,\Omega,q}\,.
\end{aligned}
$$

$\square$

**Remark 4.** *Let $M$ be a positive number such that $M_{max} \geq \max\{M_l, M_r\}$. Then, from Theorem 7, we have*

$$
|\mathfrak{C}_\omega(\mathrm{h},\mathfrak{g})| \leq \frac{M_{max}}{W}\left[\int_u^{\zeta_0} \omega(\zeta)(\zeta_0-\zeta)^p d\mu(\zeta) + \int_{\zeta_0}^v \omega(\zeta)(\zeta-\zeta_0)^p d\mu(\zeta)\right]^{1/p}
$$

$$
\times\; \|\mathfrak{g}-\overline{\mathfrak{g}}\|_{\omega,q}\; \text{for } 1 \leq p < \infty
$$

*and*

$$
|\mathfrak{C}_\omega(\mathrm{h},\mathfrak{g})| \leq \frac{M_{max}}{W}\max\{(\zeta_0-u),(v-\zeta_0)\}\|\mathfrak{g}-\overline{\mathfrak{g}}\|_{\omega,1},\; \text{for } p=\infty.
$$

Let us consider (6) for $\chi = \overline{\mathrm{h}}$. By Cauchy's inequality, we can obtain the following inequality.

**Theorem 8.** *Let $\mathrm{h},\mathfrak{g}:\Omega \in (\Omega(\mathbb{R}),\mu)$ and $\mathrm{h},\mathfrak{g},\mathrm{h}\mathfrak{g} \in L_\omega(\Omega,A,\mu)$. Then, we have*

$$
(\mathfrak{C}_\omega(\mathrm{h},\mathfrak{g}))^2 \leq \mathfrak{C}_\omega(\mathrm{h},\mathrm{h})\mathfrak{C}_\omega(\mathfrak{g},\mathfrak{g}). \tag{27}
$$

**Proof.** Applying Cauchy's inequality, we have

$$
\begin{aligned}
(\mathfrak{C}_\omega(\mathrm{h},\mathfrak{g}))^2 &= \left(\frac{1}{W}\int_\Omega \omega(\zeta)\left(\mathrm{h}(\zeta)-\overline{\mathrm{h}}\right)(\mathfrak{g}(\zeta)-\overline{\mathfrak{g}})d\mu(\zeta)\right)^2 \\
&\leq \left(\frac{1}{W}\int_\Omega \omega(\zeta)\left(\mathrm{h}(\zeta)-\overline{\mathrm{h}}\right)^2 d\mu(\zeta)\right)\left(\frac{1}{W}\int_\Omega \omega(\zeta)(\mathfrak{g}(\zeta)-\overline{\mathfrak{g}})^2 d\mu(\zeta)\right) \\
&= \mathfrak{C}_\omega(\mathrm{h},\mathrm{h})\mathfrak{C}_\omega(\mathfrak{g},\mathfrak{g}).
\end{aligned}
$$

$\square$

In contrast, a simple consequence of a Cauchy-type inequality and Sonin's identity (6) gives the following inequality.

**Theorem 9.** *Let* $h, \mathfrak{g} : \Omega \in (\Omega(\mathbb{R}), \mu)$ *and* $h, \mathfrak{g}, h\mathfrak{g} \in L_\omega(\Omega, A, \mu)$. *Then, we have*

$$(\mathfrak{C}_\omega(h, \mathfrak{g}))^2 \leq \mathfrak{C}_\omega(\mathfrak{g}, \mathfrak{g}) \frac{1}{W} \int_\Omega \omega(\zeta)(h(\zeta) - \chi)^2 d\mu(\zeta). \tag{28}$$

**Proof.** We obtain this inequality by employing Theorem 8 and Sonin's identity (6). □

The next theorem is also valid.

**Theorem 10.** *Let* $h, \mathfrak{g} \in (\Omega(\mathbb{R}), \mu)$ *and* $h, \mathfrak{g}, h\mathfrak{g} \in L_\omega(\Omega, A, \mu)$. *Then, we have*

$$\mathfrak{C}_\omega(h, h) \leq \frac{1}{W} \int_\Omega \omega(\zeta)(h(\zeta) - \chi)^2 d\mu(\zeta). \tag{29}$$

**Proof.** If we set $\mathfrak{g} = h$ in (28) we have a new inequality. □

**Remark 5.** *For* $\chi = \overline{h}$, *where* $\chi$ *is an arbitrary real number and*

$$\overline{h} = \frac{1}{W} \int_\Omega \omega(\zeta)\mathfrak{g}(\zeta) d\mu(\zeta).$$

*We have equality in* (29). *It is obvious that*

$$\mathfrak{C}_\omega(h, h) = \inf_{\alpha \in \mathbb{R}} \frac{1}{W} \int_\Omega \omega(\zeta)(h(\zeta) - \chi)^2 d\mu(\zeta) \tag{30}$$

*and we have this infimum for* $\chi = \overline{h}$.

**Theorem 11.** *Let* $h, \mathfrak{g} \in (\Omega(\mathbb{R}), \mu)$ *and* $h, \mathfrak{g}, h\mathfrak{g} \in L_\omega(\Omega, A, \mu)$, *and assume that*

$$|h(\zeta) - \chi| \leq c(\zeta) \text{ for } \mu - a.e. \ \zeta \in \Omega. \tag{31}$$

*Then, we have the inequality*

$$\mathfrak{C}_\omega(h, h) \leq \frac{1}{W} \int_\Omega \omega(\zeta)(c(\zeta))^2 d\mu(\zeta).$$

**Proof.** If (31) is valid from (29), we obtain the required inequality. □

**Theorem 12.** *With the same assumptions as in Theorem 11 and*

$$|\mathfrak{g}(\zeta) - v| \leq d(\zeta) \text{ for } \mu - a.e. \zeta \in \Omega \tag{32}$$

*is valid, where* $v$ *is a real number, we have the inequality*

$$|\mathfrak{C}_\omega(h, \mathfrak{g})| \leq \frac{1}{W} \left( \int_\Omega \omega(\zeta)(c(\zeta))^2 d\mu(\zeta) \right)^{\frac{1}{2}} \left( \int_\Omega \omega(\zeta)(d(\zeta))^2 d\mu(\zeta) \right)^{\frac{1}{2}}. \tag{33}$$

**Proof.** If (32) and (31) are valid, from (27) and (30) we obtain (33). □

**Theorem 13.** *Consider the functions* $h, \mathfrak{g}, \eta, \kappa, \gamma, \vartheta \in L^2_\omega(\Omega, A, \mu)$ *such that*
(i) $\eta + \kappa$ *and* $\gamma + \vartheta$ *are constant functions.*
(ii) $\eta(\zeta) \leq h(\zeta) \leq \kappa(\zeta)$ *and* $\gamma(\zeta) \leq \mathfrak{g}(\zeta) \leq \vartheta(\zeta)$ *for all* $\mu - a.e. \zeta \in \Omega.$

*Then, we have the inequality*

$$|\mathfrak{C}_\omega(h,\mathfrak{g})| \leq \frac{1}{4W}\|\kappa-\eta\|_{w,\Omega,2}\|\vartheta-\gamma\|_{w,\Omega,2}. \tag{34}$$

**Proof.** If $(ii)$ is valid, we have

$$\left|\mathfrak{g}(\zeta)-\frac{\gamma(\zeta)+\vartheta(\zeta)}{2}\right| \leq \frac{\vartheta(\zeta)-\gamma(\zeta)}{2}, \text{ for } \mu-a.e.\zeta \in \Omega$$

so for $v = \frac{\gamma(\zeta)+\vartheta(\zeta)}{2}$ , $d(\zeta) = \frac{\vartheta(\zeta)-\gamma(\zeta)}{2}$ and $p,q=2$ we obtain (34) from (32) and (16). □

**Corollary 4.** *Let* $h,\mathfrak{g} \in (\Omega(\mathbb{R}), \mu)$ *and* $h,\mathfrak{g},h\mathfrak{g} \in L_\omega(\Omega, A, \mu)$. *From* (28) *and* (29)*, we can obtain*

$$(\mathfrak{C}_\omega(h,\mathfrak{g}))^2 \leq \frac{1}{W^2}\int_\Omega \omega(\zeta)(h(\zeta)-\chi)^2 d\mu(\zeta)\int_\Omega \omega(\zeta)(\mathfrak{g}(\zeta)-v)^2 d\mu(\zeta). \tag{35}$$

**Remark 6.** *Substituting* $\Omega=[u,v]$, $\chi=h(\zeta_0)$ *and* $v=\mathfrak{g}(\zeta_0)$ *in* (35)*, we obtain*

$$(\mathfrak{C}_\omega(h,\mathfrak{g}))^2 \leq \frac{1}{W^2}\int_\Omega \omega(\zeta)(h(\zeta)-h(\zeta_0))^2 d\mu(\zeta)\int_\Omega \omega(\zeta)(\mathfrak{g}(\zeta)-\mathfrak{g}(\zeta_0))^2 d\mu(\zeta) \tag{36}$$

*where* $h,\mathfrak{g}:[u,v] \to \mathbb{R}$ *are differentiable on the set* $\langle u,\zeta_0\rangle \cup \langle \zeta_0,v\rangle$ *and such that* (2) *is valid and*

$$\Gamma_l = \sup_{\zeta\in\langle u,c_0\rangle} |\mathfrak{g}'(\zeta)| < \infty \text{ and } \Gamma_r = \sup_{\zeta\in\langle c_0,b\rangle} |\mathfrak{g}'(\zeta)| < \infty.$$

*Let*

$$|(h(\zeta)-h(\zeta_0))| \leq \left\{ \begin{array}{l} M_l(\zeta_0-\zeta),\ u\leq\zeta\leq\zeta_0 \\ M_r(\zeta-\zeta_0),\ \zeta_0\leq\zeta\leq v \end{array}\right.$$

*and*

$$|(\mathfrak{g}(\zeta)-\mathfrak{g}(\zeta_0))| \leq \left\{ \begin{array}{l} \Gamma_l(\zeta_0-\zeta),\ u\leq\zeta\leq\zeta_0 \\ \Gamma_r(\zeta-\zeta_0),\ \zeta_0\leq\zeta\leq v. \end{array}\right.$$

*Then, we have*

$$\int_u^v \omega(\zeta)(h(\zeta)-\eta)^2 d\mu(\zeta) = \int_u^v \omega(\zeta)(h(\zeta)-h(\zeta_0))^2 d\mu(\zeta) \tag{37}$$

$$= \int_u^{\zeta_0} \omega(\zeta)(h(\zeta)-h(\zeta_0))^2 d\mu(\zeta) + \int_{\zeta_0}^v \omega(\zeta)(h(\zeta)-h(\zeta_0))^2 d\mu(\zeta)$$

$$\leq M_l^2 \int_u^{\zeta_0} \omega(\zeta)(\zeta_0-\zeta)^2 d\mu(\zeta) + M_r^2 \int_{\zeta_0}^v \omega(\zeta)(\zeta_0-\zeta)^2 d\mu(\zeta).$$

*Similarly, we obtain*

$$\int\limits_{u}^{v} \omega(\zeta)(\mathfrak{g}(\zeta) - \kappa)^2 d\mu(\zeta) = \int\limits_{u}^{v} \omega(\zeta)(\mathfrak{g}(\zeta) - \mathfrak{g}(\zeta_0))^2 d\mu(\zeta) \tag{38}$$

$$= \int\limits_{u}^{\zeta_0} \omega(\zeta)(\mathfrak{g}(\zeta) - \mathfrak{g}(\zeta_0))^2 d\mu(\zeta) + \int\limits_{\zeta_0}^{v} \omega(\zeta)(\mathfrak{g}(\zeta) - \mathfrak{g}(\zeta_0))^2 d\mu(\zeta)$$

$$\leq \Gamma_l^2 \int\limits_{u}^{\zeta_0} \omega(\zeta)(\zeta_0 - \zeta)^2 d\mu(\zeta) + \Gamma_r^2 \int\limits_{\zeta_0}^{v} \omega(\zeta)(\zeta_0 - \zeta)^2 d\mu(\zeta).$$

*Substituting* (37) *and* (38) *in* (36), *we have*

$$(\mathfrak{C}_\omega(\mathrm{h}, \mathfrak{g}))^2 \leq \frac{1}{W^2} \left( M_l^2 \int\limits_{u}^{\zeta_0} \omega(\zeta)(\zeta_0 - \zeta)^2 d\mu(\zeta) + M_r^2 \int\limits_{\zeta_0}^{v} \omega(\zeta)(\zeta_0 - \zeta)^2 d\mu(\zeta) \right)$$

$$\times \left( \Gamma_l^2 \int\limits_{u}^{\zeta_0} \omega(\zeta)(\zeta_0 - \zeta)^2 d\mu(\zeta) + \Gamma_r^2 \int\limits_{\zeta_0}^{v} \omega(\zeta)(\zeta_0 - \zeta)^2 d\mu(\zeta) \right).$$

**Theorem 14.** *Let* $\mathrm{h} \in (\Omega(\mathbb{R}), \mu)$ *and* $\mathrm{h} \in L_\omega(\Omega, \mathcal{A}, \mu)$, $-\infty < m \leq M < +\infty$ *such that*

$$m \leq \mathrm{h}(\zeta) \leq M \text{ for } \mu - a.e. \ \zeta \in \Omega. \tag{39}$$

*Then, we have the inequality*

$$\mathfrak{C}_\omega(\mathrm{h}, \mathrm{h}) \leq \frac{(M - m)^2}{4}.$$

**Proof.** Follows from (29) and (39).  □

**Remark 7.** *If* (13) *is also valid then we have the Grüss inequality*

$$|\mathfrak{C}_\omega(\mathrm{h}, \mathfrak{g})| \leq \frac{1}{4}(M - m)(\Gamma - \gamma).$$

Finally, we state the last result of our paper, which provides an extension of Theorem 13 with more relaxed conditions.

**Theorem 15.** *Let* $\mathrm{h}, \mathfrak{g}, \eta, \kappa, \gamma, \vartheta \in L_\omega^2(\Omega, A, \mu)$ *be functions such that*
*(i)* $\eta + \kappa$ *and* $\gamma + \vartheta$ *are constant functions.*
*(ii)* $\frac{1}{W} \int\limits_{\Omega} \omega(\zeta)(\kappa(\zeta) - \mathrm{h}(\zeta))(\mathrm{h}(\zeta) - \eta(\zeta)) d\mu(\zeta) \geq 0$ *and*
$\frac{1}{W} \int\limits_{\Omega} \omega(\zeta)(\vartheta(\zeta) - \mathfrak{g}(\zeta))(\mathfrak{g}(\zeta) - \gamma(\zeta)) d\mu(\zeta) \geq 0$
*for all* $\mu - a.e. \zeta \in \Omega$. *Then, the following inequality holds.*

$$\mathfrak{C}_\omega(\mathrm{h}, \mathfrak{g}) \leq \frac{1}{4W} \|\kappa - \eta\|_{w, \Omega, 2} \|\vartheta - \gamma\|_{w, \Omega, 2} \tag{40}$$

**Proof.** From Theorem 8, we obtain

$$(\mathfrak{C}_\omega(\mathrm{h}, \mathfrak{g}))^2 \leq \mathfrak{C}_\omega(\mathrm{h}, \mathrm{h}) \mathfrak{C}_\omega(\mathfrak{g}, \mathfrak{g}). \tag{41}$$

and

$$\mathfrak{C}_\omega(\mathrm{h},\mathrm{h}) = \frac{1}{W}\int_\Omega \omega(\zeta)\mathrm{h}^2(\zeta)d\mu(\zeta) - \left(\frac{1}{W}\int_\Omega \omega(\zeta)\mathrm{h}(\zeta)d\mu(\zeta)\right)^2. \tag{42}$$

Now, by assumed conditions we have

$$\frac{1}{W}\int_\Omega \omega(\zeta)(\kappa(\zeta)-\mathrm{h}(\zeta))(\mathrm{h}(\zeta)-\eta(\zeta))d\mu(\zeta)$$
$$= \frac{1}{W}\int_\Omega \omega(\zeta)(\eta+\kappa)(\zeta)\mathrm{h}(\zeta)d\mu(\zeta) - \frac{1}{W}\int_\Omega \omega(\zeta)\mathrm{h}^2(\zeta)d\mu(\zeta)$$
$$- \frac{1}{W}\int_\Omega \omega(\zeta)\eta(\zeta)\kappa(\zeta)d\mu(\zeta) \geq 0. \tag{43}$$

Now, using (43), we obtain

$$\mathfrak{C}_\omega(\mathrm{h},\mathrm{h}) \leq \frac{1}{W}\int_\Omega \omega(\zeta)(\eta+\kappa)(\zeta)\mathrm{h}(\zeta)d\mu(\zeta) - \frac{1}{W}\int_\Omega \omega(\zeta)\eta(\zeta)\kappa(\zeta)d\mu(\zeta)$$
$$- \left(\frac{1}{W}\int_\Omega \omega(\zeta)\mathrm{h}(\zeta)d\mu(\zeta)\right)^2 - \frac{1}{4W}\|\kappa-\eta\|^2_{\omega,\Omega,2} + \frac{1}{4W}\|\kappa-\eta\|^2_{\omega,\Omega,2}. \tag{44}$$

Now, using the fact that $\eta(\zeta)+\kappa(\zeta) = \chi = \bar{\mathrm{h}}$, we can obtain

$$\chi \cdot \frac{1}{W}\int_\Omega \omega(\zeta)\mathrm{h}(\zeta)d\mu(\zeta) - \left(\frac{1}{W}\int_\Omega \omega(\zeta)\mathrm{h}(\zeta)d\mu(\zeta)\right)^2 = 0. \tag{45}$$

and

$$\frac{1}{4W}\|\kappa-\eta\|^2_{\omega,\Omega,2} + \frac{1}{W}\int_\Omega \omega(\zeta)\eta(\zeta)\kappa(\zeta)d\mu$$
$$= \frac{1}{W}\int_\Omega \omega(\zeta)\left[\left(\frac{\kappa(\zeta)-\eta(\zeta)}{2}\right)^2 + \eta(\zeta)\kappa(\zeta)\right]d\mu(\zeta)$$
$$= \frac{1}{W}\left[\int_\Omega \omega(\zeta)\left(\frac{\kappa(\zeta)+\eta(\zeta)}{2}\right)^2\right]d\mu(\zeta)$$
$$= \frac{\chi^2}{4} > 0. \tag{46}$$

Thus, (44) becomes

$$\mathfrak{C}_\omega(\mathrm{h},\mathrm{h}) \leq \frac{1}{4W}\|\kappa-\eta\|^2_{\omega,\Omega,2} - \frac{\chi^2}{4}$$
$$\leq \frac{1}{4W}\|\kappa-\eta\|^2_{\omega,\Omega,2}. \tag{47}$$

In the similar passion, one can obtain

$$\mathfrak{C}_\omega(\mathfrak{g},\mathfrak{g}) \leq \frac{1}{4W}\|\Gamma-\gamma\|^2_{\omega,\Omega,2}. \tag{48}$$

Substituting bounds (47) and (48) into (41), we establish (40), which completes the proof. □

### 3. Applications to new Jensen–Grüss Bounds

We start with the weighted version of Korkine's identity [7] (p. 242) (see also [26]) :

$$\mathfrak{C}_\omega(Y, h) = \frac{1}{2W^2} \int_\Omega \int_\Omega \omega(\zeta)\omega(y)\Big(Y(\zeta) - Y(y)\Big)\Big(h(\zeta) - h(y)\Big) d\mu(\zeta) d\mu(y) \tag{49}$$

which holds for all $\mu$-measurable functions $\omega, Y, h$, with $\omega(\zeta) > 0$ for $\mu - a.e.$ $\zeta \in \Omega$, such that $Y, h \in L_\omega(\Omega, \mathcal{A}, \mu)$.

**Theorem 16.** *Let* $Y : I \subset \mathbb{R} \to \mathbb{R}$ *be a differentiable mapping with a continuous first derivative. Let* $h : \Omega \to I$ *such that* $h, \eta, \kappa, Y \circ h, Y' \circ h \in L^2_\omega(\Omega, A, \mu)$ *and* $h, \eta, \kappa$ *satisfy the following conditions:*

*(i)* $\eta + \kappa$ *is a constant function.*

*(ii)* $\frac{1}{W} \int_\Omega \omega(x)(\kappa(\zeta) - h(\zeta))(h(\zeta) - \eta(\zeta)) d\mu(\zeta) \geq 0$ *for all* $\mu - a.e.\zeta \in \Omega$.

*Furthermore, suppose that there exists* $m, M \in \mathbb{R}$ *satisfying*

$$m \leq Y'(\zeta) \leq M, \qquad \text{for all} \quad \zeta \in I. \tag{50}$$

*Then, we have the following inequalities*

$$\left| \frac{1}{W} \int_\Omega \omega(\zeta)(Y \circ h)(\zeta) d\mu(\zeta) - Y\left(\frac{1}{W} \int_\Omega \omega(\zeta)h(\zeta) d\mu(\zeta)\right) \right| \tag{51}$$

$$\leq \frac{(M-m)}{2} \left( \frac{1}{W} \int_\Omega \omega(\zeta)h^2(\zeta) d\mu(\zeta) - \left(\frac{1}{W} \int_\Omega \omega(\zeta)h(\zeta) d\mu(\zeta)\right)^2 \right)^{\frac{1}{2}}$$

$$\leq \frac{(M-m)}{4\sqrt{W}} \|\kappa - \eta\|_{\omega,\Omega,2}.$$

**Proof.** As a consequence of the mean value theorem, for the points $\zeta, y \in I$, we can write that there exists $\mathfrak{z}$, $\zeta \leq \mathfrak{z} \leq y$ such that

$$Y(\zeta) - Y(y) = Y'(\mathfrak{z})(\zeta - y) \tag{52}$$

Using (52) for $\zeta = \overline{h} = \frac{1}{W} \int_\Omega \omega(\zeta)h(\zeta) d\mu(\zeta)$ and $y = h$, we conclude that there exists $g$ $(\overline{h} \leq g \leq h)$ such that

$$Y\left(\overline{h}\right) - Y(h) = Y'(g)\left(\overline{h} - h\right) \tag{53}$$

Now, multiplying (53) by $\omega(\zeta)$ and integrating over $\Omega$ yields

$$WY\left(\overline{h}\right) - \int_\Omega \omega(\zeta)Y(h(\zeta) d\mu(\zeta) = \overline{h} \int_\Omega \omega(\zeta)Y'(g(\zeta)) d\mu(\zeta) - \int_\Omega \omega(\zeta)Y'(g(\zeta))h(\zeta) d\mu(\zeta).$$

Dividing by $W$, we obtain

$$\frac{1}{W} \int_\Omega \omega(\zeta)(Y \circ h)(\zeta) d\mu(\zeta) - Y\left(\frac{1}{W} \int_\Omega \omega(\zeta)h(\zeta) d\mu(\zeta)\right)$$

$$= \frac{1}{W} \int_\Omega \omega(\zeta)Y'(g(\zeta))h(\zeta) d\mu(\zeta) - \frac{1}{W} \int_\Omega \omega(\zeta)h(\zeta) d\mu(\zeta) \frac{1}{W} \int_\Omega \omega(\zeta)Y'(g(\zeta)) d\mu(\zeta). \tag{54}$$

Now, taking the modulus on both sides and using the weighted Krokine's identity (49) gives

$$
\left| \frac{1}{W} \int_\Omega \omega(\zeta)(Y \circ h)(\zeta) d\mu(\zeta) - Y\left( \frac{1}{W} \int_\Omega \omega(\zeta) h(\zeta) d\mu(\zeta) \right) \right|
$$

$$
= \left| \frac{1}{W} \int_\Omega \omega(\zeta) Y'(g(\zeta)) h(\zeta) d\mu(\zeta) - \frac{1}{W} \int_\Omega \omega(\zeta) h(\zeta) d\mu(\zeta) \frac{1}{W} \int_\Omega \omega(\zeta) Y'(g(\zeta)) d\mu(\zeta) \right|
$$

$$
= \left| \mathfrak{C}_\omega(h, Y'(g)) \right| \le \frac{1}{2W^2} \int_\Omega \int_\Omega \omega(\zeta) \omega(y) |h(\zeta) - h(y)| \left| Y'(g(\zeta)) - Y'(g(y)) \right| d\mu(\zeta) d\mu(y).
$$

Now, applying a Cauchy–Schwartz inequality, we can state that the last expression is less than

$$
\left| \frac{1}{W} \int_\Omega \omega(\zeta)(Y \circ h)(\zeta) d\mu(\zeta) - Y\left( \frac{1}{W} \int_\Omega \omega(\zeta) h(\zeta) d\mu(\zeta) \right) \right|
$$

$$
\le \mathfrak{C}_\omega^{\frac{1}{2}}(h, h) \ \ \mathfrak{C}_\omega^{\frac{1}{2}}(Y'(g), Y'(g)). \tag{55}
$$

Now, utilizing weighted Grüss inequality on second term, we obtain

$$
\le \mathfrak{C}_\omega^{\frac{1}{2}}(h, h) \ \ \frac{1}{2}(M - m)
$$

$$
= \left( \frac{1}{W} \int_\Omega \omega(\zeta) h^2(\zeta) d\mu(\zeta) - \left( \frac{1}{W} \int_\Omega \omega(\zeta) h(\zeta) d\mu(\zeta) \right)^2 \right)^{\frac{1}{2}} \frac{(M - m)}{2}.
$$

Now, utilizing Theorem 15 for $Y = h$ on first term, we obtain

$$
\le \frac{1}{2\sqrt{W}} \| \kappa - \eta \|_{\omega, \Omega, 2} \frac{(M - m)}{2}.
$$

$\square$

Now, we provide refinements of Jensen–Grüss inequality for functions $h, \eta, \kappa \in L_w^2(\Omega, A, \mu)$ satisfying the conditions assumed in Theorem 13.

**Corollary 5.** *Under the assumptions of Theorem 16, if* $h, \eta, \kappa$ *satisfy the conditions*

(i)　$\eta + \kappa$ *is a constant function.*

(ii)　$\eta(\zeta) \le h(\zeta) \le \kappa(\zeta)$ *for all* $\mu - a.e. \zeta \in \Omega$.

　　*then, there exist* $m, M \in \mathbb{R}$ *such that* (50) *is valid, we again obtain inequalities given in* (51).

**Proof.** Similar to that of Theorem 16. $\square$

Now, we use other results of the paper to provide new Jensen–Grüss inequalities.

**Theorem 17.** *Let* $\Upsilon$ *be as defined in Theorem 16 and* $h : \Omega \to I$ *be a function such that* $h, \Upsilon \circ h, \Upsilon' \circ h \in L_\omega(\Omega, A, \mu)$ *for,* $\mu - a.e. \zeta \in \Omega$. *Furthermore, there exist* $m, M \in \mathbb{R}$ *satisfying* (50). *Then, we have the following inequalities*

$$
\left| \frac{1}{W} \int_\Omega \omega(\zeta)(\Upsilon \circ h)(\zeta) d\mu(\zeta) - \Upsilon \left( \frac{1}{W} \int_\Omega \omega(\zeta) h(\zeta) d\mu(\zeta) \right) \right| \tag{56}
$$

$$
\leq \frac{(M-m)}{2} \left( \frac{1}{W} \int_\Omega \omega(\zeta) h^2(\zeta) d\mu(\zeta) - \left( \frac{1}{W} \int_\Omega \omega(\zeta) h(\zeta) d\mu(\zeta) \right)^2 \right)^{\frac{1}{2}}
$$

$$
\leq \frac{(M-m)}{2} \left( \frac{1}{W} \int_\Omega \omega(\zeta)(h(\zeta) - \chi)^2 d\mu(\zeta) \right)^{\frac{1}{2}}.
$$

**Proof.** We have already established in the proof of Theorem 16 that

$$
\left| \frac{1}{W} \int_\Omega \omega(\zeta)(\Upsilon \circ h)(\zeta) d\mu(\zeta) - \Upsilon \left( \frac{1}{W} \int_\Omega \omega(\zeta) h(\zeta) d\mu(\zeta) \right) \right|
$$

$$
\leq \mathfrak{C}_\omega^{\frac{1}{2}}(h, h) \ \ \mathfrak{C}_\omega^{\frac{1}{2}}(\Upsilon'(g), \Upsilon'(g))
$$

$$
\leq \mathfrak{C}_\omega^{\frac{1}{2}}(h, h) \ \ \frac{1}{2}(M-m)
$$

$$
= \left( \frac{1}{W} \int_\Omega \omega(\zeta) h^2(\zeta) d\mu(\zeta) - \left( \frac{1}{W} \int_\Omega \omega(\zeta) h(\zeta) d\mu(\zeta) \right)^2 \right)^{\frac{1}{2}} \frac{(M-m)}{2}. \tag{57}
$$

Now, utilizing Theorem 10 on the first term, we obtain the required results. $\square$

The next result is the direct consequence of Theorem 11.

**Corollary 6.** *Under the assumptions of Theorem 17, if*

$$
|h(\zeta) - \chi| \leq c(\zeta) \text{ for } \mu - a.e. \ \zeta \in \Omega. \tag{58}
$$

*Then, we have the following inequalities*

$$
\left| \frac{1}{W} \int_\Omega \omega(\zeta)(\Upsilon \circ h)(\zeta) d\mu(\zeta) - \Upsilon \left( \frac{1}{W} \int_\Omega \omega(\zeta) h(\zeta) d\mu(\zeta) \right) \right| \tag{59}
$$

$$
\leq \frac{(M-m)}{2} \left( \frac{1}{W} \int_\Omega \omega(\zeta) h^2(\zeta) d\mu(\zeta) - \left( \frac{1}{W} \int_\Omega \omega(\zeta) h(\zeta) d\mu(\zeta) \right)^2 \right)^{\frac{1}{2}}
$$

$$
\leq \frac{(M-m)}{2} \left( \frac{1}{W} \int_\Omega \omega(\zeta) \left( c(\zeta) \right)^2 d\mu(\zeta) \right)^{\frac{1}{2}}.
$$

The next result is an important consequence of Corollary 4 and Remark 6.

**Corollary 7.** *Let* $\Upsilon : I \subset \mathbb{R} \to \mathbb{R}$ *be a differentiable mapping with a continuous first derivative. Furthermore, let* $h : [u, v] \to I$ *be differentiable on the set* $(u, \zeta_0) \cup (\zeta_0, v)$ *such that* (1.7) *is valid and*

$$
|(h(\zeta) - h(\zeta_0))| \leq \begin{cases} M_l(\zeta_0 - \zeta), & u \leq \zeta \leq \zeta_0 \\ M_r(\zeta - \zeta_0), & \zeta_0 \leq \zeta \leq v. \end{cases}
$$

*Furthermore, suppose that* $\mathrm{h}, \mathrm{Y} \circ \mathrm{h}, \mathrm{Y}' \circ \mathrm{h}, \in L_{\omega,[u,v]}$ *and there exists an* $m, M \in \mathbb{R}$ *such that* (50) *holds. Then, we have the following inequalities*

$$
\left| \frac{1}{W} \int_u^v \omega(\zeta)(\mathrm{Y} \circ \mathrm{h})(\zeta)d\mu(\zeta) - \mathrm{Y}\left( \frac{1}{W} \int_u^v \omega(\zeta)\mathrm{h}(\zeta)d\mu(\zeta) \right) \right| \tag{60}
$$

$$
\leq \frac{(M-m)}{2} \left( \frac{1}{W} \int_u^v \omega(\zeta)\mathrm{h}^2(\zeta)d\mu(\zeta) - \left( \frac{1}{W} \int_u^v \omega(\zeta)\mathrm{h}(\zeta)d\mu(\zeta) \right)^2 \right)^{\frac{1}{2}}
$$

$$
\leq \frac{(M-m)}{2} \left( \frac{1}{W} \left( M_l^2 \int_u^{\zeta_0} \omega(\zeta)(\zeta_0 - \zeta)^2 d\mu(\zeta) + M_r^2 \int_{\zeta_0}^v \omega(\zeta)(\zeta_0 - \zeta)^2 d\mu(\zeta) \right) \right)^{\frac{1}{2}}.
$$

**Proof.** Using Corollary 4 and employing Remark 6 by substituting $\Omega = [u,v]$, $\chi = \mathrm{h}(x_0)$, we obtain

$$
\mathfrak{C}_\omega(\mathrm{h}, \mathrm{h}) \leq \frac{1}{W} \left( M_l^2 \int_u^{\zeta_0} \omega(\zeta)(\zeta_0 - \zeta)^2 d\mu(\zeta) + M_r^2 \int_{\zeta_0}^v \omega(\zeta)(\zeta_0 - x)^2 d\mu(\zeta) \right).
$$

Now, considering this bound for $\mathfrak{C}_\omega(\mathrm{h}, \mathrm{h})$ in the first term of (57), we will obtain the required results. $\square$

## 4. Conclusions

In the present article, we introduced generalizations of pre-Grüss- and Grüss-type inequalities in weighted Lebesgue $L_\omega(\Omega, \mathcal{A}, \mu)$ spaces by utilizing the weighted Sonin's identity. In the newly generalized inequalities, the bounding constants are improved with bounded functions in weighted Lebesgue $L_\omega^2(\Omega, \mathcal{A}, \mu)$ spaces. Weighted bounds for the Čebyšev functional as well as the weighted Sokolov's inequality are established. Several special and interested cases are presented as well. We also proved the above generalizations by employing weaker assumptions. Finally, we use our obtained results to construct new refinements and bounds for Jensen–Grüss type inequalities. It is pertinent to mention that such results can be discussed for weighted sequence spaces using the discrete Čebyšev- and Grüss-type inequalities. In the future, it is possible to expand on the results of this study by considering isotonic linear functionals or inner product spaces.

**Author Contributions:** Conceptualization, J.P. and S.I.B.; methodology, S.T.-S.;validation, S.T.-S. and S.I.B.; investigation, S.T.-S.; writing—original draft preparation, S.T.-S. and S.I.B.; writing—review and editing, S.T.-S. and S.I.B.; visualization, S.I.B.; supervision, J.P.; project administration, J.P.; funding acquisition, S.T.-S. All authors have read and agreed to the published version of the manuscript.

**Funding:** This work was funded in part by University of Split, Faculty of Chemistry and Technology, Croatia.

**Institutional Review Board Statement:** Not applicable.

**Informed Consent Statement:** Not applicable.

**Data Availability Statement:** Not applicable.

**Acknowledgments:** All the authors are thankful to their respective institutes.

**Conflicts of Interest:** The authors declare no conflict of interest.

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
