# Peer review of "Generalized Čebyšev and Grüss Type Results in Weighted Lebesgue Spaces"

_mathematics, doi:10.3390/math11071756_

Round 1

Author Response

In our revised version we have done all  the suggestions and minor essential revisions :

Point 1: page 2, It must be variable ζ instead of z in the definition of norm defined for L∞ [u,v] that is ||h||∞ = ess sup z∈[u,v]   

Response 1:  Yes, it must variable ζ instead of z in the definition of norm. We corrected it.

Point 2: In page 3, Put comma after the end of this line i.e.: It is interesting to note that as a special case p = q = 2.

Response 2: Thank you for remark. It should be comma there.

Point 3: In page 5, Proof of Theorem 5. Put comma(,) before we get (12) from (9).

Response 3: Thank you for remark. It should be comma there.

Point 4: In page 6, Corollary 1, elaborate what are χi?.

Response 4 : χi are  arbitrary real numbers.

Point 5: In page 11, Remark 4: There are too many M above already so keep things clear and precise you may denote it with Mmax or some different noataion etc.

Response 5: We agree with it. We put Mmax instead of M.

Point 6:  A conclusion section can be added with some future directions and goals as well.

Response 6 : We agree, so we added a conclusion in which we mentioned some future goals.

Reviewer 2 Report

The aim of the paper is to give generalization of the Gr\"uss inequality in a weighted space. The paper is well elaborated. It contains a great number of refined results. In conclusion we recommend the publication. We have only some few remarks: \begin{itemize} \item Page 2, line 31: The symbol $\xi$ is not defined and the symbol $\Gamma$ does not appear. \item Page 4, the last formula in line 57, i.e. which is named the identity of Sonin is unclear. Who is function $w$? Because number $\chi$ is arbitrary it is necessary to have $$\int_u^v\omega(\zeta)\left(g(\zeta)-\int_u^vw(\zeta)g(\zeta)d\zeta\right)d\mu(\zeta)=0.$$ This equality is unclear. We do not understand why formula (6) is a generalization of the Sonin identity, because in Sonin's identity the factor $\frac1W$ does not appear. \item Page 5, line 73: we get (12) from (10). \item Page 6, the first line (line 81) we have $\le$, instead of $=$.

\end{itemize}

Author Response

In our revised version we have done all  the suggestions and minor essential revisions :

Point 1:  Page 2, line 31: The symbol ξ is not defined and the symbol Γ does not appear.

Response 1: We agree with this remark. We have corrected everything as follows: There should be ξ instead of and and it should be written there “for a.e. ξ ∈ [u, v]”. We also deleted the symbols Γ and γ  because they should not appear.

Point 2:  Page 4, the last formula in line 57, i.e. which is named the identity of Sonin is unclear. Who is function w? Because number χ is arbitrary it is necessary to have Z v u ω(ζ) g(ζ) − Z v u w(ζ)g(ζ)dζ dµ(ζ) = 0. This equality is unclear. We do not understand why formula (6) is a generalization of the Sonin identity, because in Sonin’s identity the factor 1 W does not appear.

Response 2: Thank you for your comment. In fact, their should be $\omega$ instead of w in the line 57 Sonin’s formula. Also weight function is normalized weight function that is W=1 there.  So we mention it now clearly . Rest is correct as Formula (6)  is a generalization as integrals are over general weighted  Lebesgue (Ω, A, µ) measurable space (and not integral from u to v) and factor 1/W in the last  formula in line 57 should not be their as their the weight function is normalized function so W=1 there. 

Point 3: Page 5, line 73: we get (12) from (10).

Response 3: Yes, we agree. We corrected it

Point 4:  Page 6, the first line (line 81) we have ≤, instead of =.

Response 4: It should be ≤, instead of =. We corrected it

Reviewer 3 Report

In this paper, the authors give generalizations of the Sokolov and Grüss type inequalities in weighted Lebesgue spaces. The obtained results are used to determine bounds for Jensen-Grüss type differences.

The paper contains many results that are improvements of known inequalities. The proofs are concise, but clear. The steps of reasoning are provided with precise short comments. 

I think it would be useful to add a conlusion that will allow the reader to navigate more easily among the many results presented in the article.

supplementative report:

The problem considered in this work is to obtain new inequalities of the Sokolov and Grüss type in weighted Lebesgue spaces. The authors showed many inequalities of this type, which enriches the subject of this research area. They also apply the obtained results to determine bounds for Jensen-Grüss type differences. The proofs are clear, the steps of reasoning are accompanied by precise short comments.
The article provides a sufficient introduction presenting the known results on the topic under study. The references are appropriate. In the main part of the article, many inequalities are given, but the interrelationships and differences between them are not explicitly described. I think that in order not to unnecessarily change the structure of the article, comments on the relationships between the presented results can be placed in the conclusion. So I suggest adding the conclusion that currently doesn't exist.

Author Response

In our revised version we have done all  the suggestions and minor essential revisions :

Point 1:  The article provides a sufficient introduction presenting the known results on the topic under study. The references are appropriate. In the main part of the article, many inequalities are given, but the interrelationships and differences between them are not explicitly described. I think that in order not to unnecessarily change the structure of the article, comments on the relationships between the presented results can be placed in the conclusion. So I suggest adding the conclusion that currently doesn't exist.

Response 1: We have added the conclusion and some future direction as well.